# Flavonoid Production: Current Trends in Plant Metabolic Engineering and De Novo Microbial Production

**DOI:** 10.3390/metabo13010124

**Published:** 2023-01-13

**Authors:** Hasnat Tariq, Saaim Asif, Anisa Andleeb, Christophe Hano, Bilal Haider Abbasi

**Affiliations:** 1Department of Biotechnology, Quaid-i-Azam University, Islamabad 45320, Pakistan; 2Department of Biosciences, COMSATS University, Islamabad 45550, Pakistan; 3Laboratoire de Biologie des Ligneux et des Grandes Cultures (LBLGC), INRAE USC1328, Eure et Loir Campus, Université d’Orléans, 28000 Chartres, France

**Keywords:** flavonoids, biosynthesis, metabolic engineering, microbial production, metabolic pathways, synthetic biology, co-culture engineering

## Abstract

Flavonoids are secondary metabolites that represent a heterogeneous family of plant polyphenolic compounds. Recent research has determined that the health benefits of fruits and vegetables, as well as the therapeutic potential of medicinal plants, are based on the presence of various bioactive natural products, including a high proportion of flavonoids. With current trends in plant metabolite research, flavonoids have become the center of attention due to their significant bioactivity associated with anti-cancer, antioxidant, anti-inflammatory, and anti-microbial activities. However, the use of traditional approaches, widely associated with the production of flavonoids, including plant extraction and chemical synthesis, has not been able to establish a scalable route for large-scale production on an industrial level. The renovation of biosynthetic pathways in plants and industrially significant microbes using advanced genetic engineering tools offers substantial promise for the exploration and scalable production of flavonoids. Recently, the co-culture engineering approach has emerged to prevail over the constraints and limitations of the conventional monoculture approach by harnessing the power of two or more strains of engineered microbes to reconstruct the target biosynthetic pathway. In this review, current perspectives on the biosynthesis and metabolic engineering of flavonoids in plants have been summarized. Special emphasis is placed on the most recent developments in the microbial production of major classes of flavonoids. Finally, we describe the recent achievements in genetic engineering for the combinatorial biosynthesis of flavonoids by reconstructing synthesis pathways in microorganisms via a co-culture strategy to obtain high amounts of specific bioactive compounds

## 1. Introduction

Flavonoids comprise an essential and diverse class of polyphenolic compounds that are synthesized in plants as bioactive secondary metabolites, consisting of more than 6000 classified phenolics [1,2,3,4]. The synthesis of these secondary metabolites in plants is tissue-specific and highly regulated [5]. They contain 15 carbon atoms and two phenyl rings linked together with a 3-carbonated heterocyclic ring, forming a C6-C3-C6 structure [6,7,8,9]. Due to variations in the structure of the backbone, an enormous level of chemical diversity is observed in flavonoids [10]. Based on these variations, the flavonoids are organized into six main groups, namely: flavan-3-ols, flavanones, flavones, flavanols, anthocyanidins, and isoflavones [1,4,6,10]. These natural compounds are generally found in citrus grain seeds, vegetables, fruits, nuts, chocolate, cocoa, legumes, black tea, red wine, green tea, soy, herbs, onions, berries, apple, grapes, and cider [3,6,7,8,11]. In plants, flavonoids are responsible for protecting against oxidative stress, by acting as free radical scavengers, and provide coloration to flowers, leaves, and fruits, hence attracting pollinators. Moreover, these phytochemicals act as UV filters and protect plants from damage caused by UV radiation, function as signaling molecules, serve as antimicrobial defensive compounds, and act as chelators for toxic heavy metals [6,10,12,13]. Flavonoid compounds have been reported to have a wide range of potential biological applications in humans, including antibacterial [14], antifungal [8], antiviral [15], anticancer [1], cardioprotective [6], anti-inflammatory [16], antidiabetic [3], antiaging [17], and radioprotective activities [9].

The current methods of producing flavonoids are largely dependent on plant extraction, which has several limitations and drawbacks. Firstly, many plants rich in flavonoids also act as a food source. Hence, their consumption for the production of flavonoids may result in depletion of the overall food supply [18,19]. Secondly, extraction yields are limited due to low concentrations or loses during extraction; moreover, long breeding seasons, unfavorable climate conditions, small plantation sizes, cultivation difficulties, and variation in plant species also effect production [5,18,20]. Simultaneously, the flavonoid concentration in these plants is comparatively low, limiting their large-scale production [18]. An alternative approach could be chemical synthesis, but due to the complexity of some flavonoid structures, this synthesis approach is expensive and time-consuming [18]. Moreover, the use of potentially hazardous chemicals and intense reaction conditions also restrict the de novo synthesis of these plant secondary metabolic compounds [5,20,21]. As a result, these factors add up to a costly, energy-intensive preparation process with room for further improvement as well [18]. In this context, the reconstitution of biosynthetic gene fragments in plants and industrially important microorganisms holds a lot of promise for flavonoid exploration and scalable production [11,22]. Plant metabolic engineering promises to increase the production of specific beneficial flavonoids and to expand the chemical range of novel flavonoids [11,23], but still, it is difficult and technically challenging to engineer a plant genome. The editing of a plant genome for enhanced metabolite production often results in several challenges and limitations in terms of titer variability, aggregation susceptibility, growth rate, stress, and heterogeneity of culture [24]. The various methods of synthesizing flavonoids are depicted in Figure 1.

Recently, the microbial production of flavonoids has received ample attention due to its low energy requirements, high purity of the product, and low emissions of waste, such as sulfates, nitrates, or nitrites [20]. Synthetic and system biology technologies have revolutionized the field of metabolic engineering over the past two decades, allowing for the establishment of bio-based production in numerous engineered microbes [25,26]. This approach has several benefits, such as a rapid growth rate, safety, economic substrates, ease of culture, range of various genetic techniques, and high concentration of desired metabolites as compared to the native host [24]. In recent years, apart from *Escherichia coli*, a wide array of host strains, including *Lactococcus lactis*, *Yarrowia lipolytica*, and *Saccharomyces cerevisiae*, have been tested for flavonoid production [11]. Nowadays, a different approach involving modular co-culture engineering has surfaced to overcome the constraints of the conventional monoculture approach by harnessing the power of two or more strains of engineered microbes to reconstruct the target biosynthetic pathway. In this way, it has largely reduced the biosynthesis labor, as well as the associated metabolic burden on each microbial strain [27]. In this review, we aim to discuss recent trends in the biosynthesis and metabolic engineering of flavonoids in plants. We have also discussed the microbial production of pharmaceutically and industrially important flavonoids utilizing both monoculture and co-culture approaches.

## 2. Flavonoid Biosynthetic Pathways in Plants

There are two distant biosynthetic pathways in plants for the production of flavonoid compounds, e.g., the acetate pathway and the shikimic acid pathway. Initially, the acetate pathway that forms ring A (of flavonoids) and the shikimate pathway that forms ring B (of flavonoids) merge together through a linking chain called ring C (generating the C6-C3-C6 structural components or building-blocks of the flavonoid skeleton) leading to the biosynthesis of flavonoids. The transformation of glucose leads to the synthesis of three molecules of malonyl-CoA from which ring A is synthesized, whereas the shikimate pathway generates the phenyl propanoids through the production of phenylalanine, which gives 4-coumaroyl-CoA molecules as a ring B.

The shikimate pathway in plants leads to the biosynthesis of phenylpropanoids via the phenylpropanoid pathway. Phenylalanine, which comes from the shikimate pathway, acts as a precursor amino acid in the phenylpropanoid biosynthesis pathway, leading to the synthesis of flavonoids. This pathway is unique to plants because in addition to flavonoid biosynthesis, it also produces coumarins, esters, lignin, hydroxycinnamates, and other plant secondary metabolites [28]. Flavonoids are vital for plant survival and play an essential role in plant metabolism, especially in defense and pigmentation [29]. The phenylpropanoid production occurs in specialized plant tissues and cells; however, biosynthesis may also take place in response to environmental factors, such as ultraviolet (UV) damage or pathogenic attacks [30]. Naringenin chalcone also acts as a precursor molecule to produce flavonoids via the phenylpropanoid pathway [31]. The first step involved in the phenylpropanoid pathway is the phenylalanine ammonia-lyase (PAL)-mediated deamination of the amino acid phenylalanine to cinnamic acid. Next, cinnamate-4-hydroxylase (C4H) hydroxylates cinnamic acid to *p*-coumaric acid, which is later activated by para-coumarate-CoA ligase (PCL) to form *p*-coumaroyl-CoA. The step-by-step condensation of the malonyl-CoA three acetate units with *p*-coumaroyl-CoA is catalyzed by chalcone synthase (CHS), which ultimately results in the production of naringenin chalcone [32]. In the final step of flavonoid biosynthesis, chalcone isomerase (CHI) converts naringenin chalcone to flavanone or naringenin, as represented in Figure 2. Different flavonoids, such as flavones, isoflavones, and anthocyanidins, are further generated from various structural modifications of naringenin [33]. Due to the specific action of CHS, a variety of secondary metabolites are produced as part of the phenylpropanoid pathway [34,35]. The A and B phenyl rings, which form the backbone of the flavonoid structure, arise via the action of CHS, as it combines three molecules of malonyl-CoA with *p*-coumaroyl-CoA to form a chalcone [36]. CHI takes part in the synthesis of the important C ring of the flavonoid backbone resulting in the formation of flavanone (naringenin). Isoflavone synthase (IFS) catalyzes the transition of flavanone to isoflavone. Flavanone 3 β-hydroxylase (F3H) hydroxylates naringenin to form dihydroflavonol, which is then converted to leucoanthocyanidin by dihydroflavonol 4-reductase (DFR) [37]. Further catalysis mediated by anthocyanidin synthase (ANS) leads to the formation of anthocyanidin from leucoanthocyanidins, whereas leucoanthocyanidin reductase (LAR) undergoes the reduction of leucoanthocyanidins to flavan-3-ol (also referred to as flavanols, a complex class of flavonoids) [38].

Even though there have been numerous advancements in technologies in recent years, there are still challenges in engineering flavonoid pathways in plants. Some of the major hurdles include the yield (very low metabolite production or resource constraints of sufficient biomass), purity (heterogenous mixture of several molecules making it difficult to identify one substance), and extraction (degradation of chemical structure). Apart from these, regulation at molecular, enzymatic, and cellular levels is also associated with some key obstacles to genetic engineering approaches of metabolic pathways [22].

## 3. An Overview of Key Enzymes Involved in Biosynthesis of Flavonoids

The flavonoid biosynthetic reactions are regulated by several key enzymes present in various plant species. The identification of numerous enzymes associated with flavonoid synthesis is the result of collective efforts of different researchers and extensive genetic research performed in this area, over the past few decades [32]. The pathways involving enzymes are potential targets for metabolic engineering, to perform genetic modifications in order to increase drought and pest resistance, improve yields, and produce hybrid plants having higher nutritional value with desired characteristics [39]. There are hundreds of enzymes involved in flavonoid biosynthesis in different plant species, but we have described only key metabolic enzymes of biosynthesis pathways here. Some major enzymes and their roles in flavonoid biosynthetic pathways have been discussed in Table 1. Enzymes that participate in plant primary metabolism are also correlated with the production of secondary metabolites, as is the case with flavonoid biosynthesis [40]. CHI and CHS, which are associated with the synthesis of the basic three-ring flavonoid backbone, are also involved in fatty acid metabolism in plants. In plants, many of the enzymes involved in the production of primary and secondary metabolites, such as flavonoid biosynthesis, belong to the dehydrogenase/reductase and 2-oxoglutarate-dependent dehydrogenase (2OGD) enzyme families. F3H, CHI, and CHS are the major enzymes involved in the initial flavonoid synthesis in many plants. Flavonoids have defensive properties and thus serve as part of the survival mechanism present in many plant species. CHS can be found in numerous plant species ranging from angiosperms to bryophytes and is also present in some bacteria, fungi, and microalgae [41]. CHS belongs to the type III polyketide synthase (PKS) superfamily and takes part in the polyketide formation of flavonoid biosynthesis [42]. The presence of such versatile secondary metabolites in plants may be due to an abundance of PKSs. Like CHS, stilbene synthase (STS) is also classified as a member of the type III PKSs superfamily. STSs and CHSs specifically catalyze various cyclization reactions to generate different products. Like CHS, CHI is also part of the fatty acid metabolism pathway in plants, involved in flavonoid biosynthesis, and is present in numerous plant species, as well as fungi and bacteria [43]. CHI is classified into two types, type I and type II CHI, and both are involved in flavonoid biosynthesis and carry out isomerization reactions resulting in naringenin production [44]. The enhanced expression of the CHI gene leads to an increase in flavonoid production in plants.

Enzymes, such as flavonol synthase (FLS), flavone synthase (FNS), F3H, and ANS/LDOX, belong to the 2OGD superfamily of proteins present in the kingdom Plantae [45]. The 2OGD action starts by incorporating the substrate molecule along with oxoglutarate (2OG) and O_2_, which upon catalysis, results in the formation of oxidized products, as well as CO_2_ and succinate (RH + 2OG + O_2_ → ROH + succinate + CO_2_) [46]. 2OGDs are associated with numerous oxidation reactions and play an intricate part in plant secondary metabolite production, including terpenoids, alkaloids, and flavonoids [47,48]. The cytochrome P450 (CYP) protein family includes enzymes, such as flavonoid 3′5 hydroxylase (F3′5′H), flavonoid 3′-hydroxylase (F3′H), flavanone 2-hydroxylase (F2H), flavone synthase II (FNS II), and IFS. CYPs are heme-containing membrane-bound proteins located on the surface of the endoplasmic reticulum (ER) and are present in numerous species in the plant kingdom. They are involved in the production of primary, but also secondary plant metabolites, by carrying out hydroxylation reactions [49]. F3′5′H and F3′H activity results in the fabrication of anthocyanins [50], whereas F2H, FNS II, and IFS are associated with the biosynthesis of isoflavones and flavones [51]. ANR and LAR, which are part of the short-chain dehydrogenase/reductase (SDR) family, are present abundantly in eukaryotes, prokaryotes, archaea, and viruses. SDRs contribute to the enhanced production of secondary plant metabolites, such as flavonoids, alkaloids, and terpenoids [52], and are also engaged in the primary metabolism of carbohydrates, hormones, and lipids. ANR and LAR are both involved in the biosynthesis of anthocyanin. ANR takes part in the synthesis of epicatechin from anthocyanidin, whereas LAR catalyzes the formation of leucoanthocyanidin to catechin [53]. Enzymes, such as CHI and IFS, take part in isomerization reactions [54], whereas MT and AT catalyze the transfer of an R-group between molecules performing transferase activity [55].

**Table 1 metabolites-13-00124-t001:** Enzymes Associated with Biosynthesis of Flavonoids.

Enzyme	E.C Number	Abbreviation	Class	Family	Function	References
Phenylalanine ammonia lyase	4.3.1.24	PAL	Lyases	Amino acid lyases	Catalyzes the non-oxidative deamination of L-phenylalanine and L-tyrosine	[56]
Cinnamate-4-hydroxylase	1.14.14.91	C4H	Oxidoreductases	Cytochrome P450 (CYP)	Hydroxylation of cinnamic acid	[57]
Para-coumarate-CoA ligase	6.2.1.12	PCL	Ligases	Adenylate formers	Phenylpropanoid metabolism for secondary compound synthesis	[58]
Chalcone synthase	2.3.1.74	CHS	Transferases	Polyketide Formers	Catalytic conversion of coumaroyl-CoA and malonyl-CoA	[59]
Chalcone isomerase	5.5.1.6	CHI	Isomerases	Not specified	Catalyzing the stereospecific isomerization of chalcone	[60]
Chalcone reductase	1.1.1.-	CHR	Oxidoreductases	Aldo/Keto formers	Formation of chalcones	[53]
Isoflavone synthase	5.4.99.-	IFS	Isomerases	CYP	Catalyzes 2,3 aryl ring migration of flavanones	[61]
Isoflavone reductase	1.3.1.45	IFR	Oxidoreductases	NADPH reductases	Synthesis of glyceollins from daidzein	[62]
Flavone synthase	1.14.11.22	FNS	Oxidoreductases	Dioxygenases	Catalyzes a double bond formation between C2 and C3 of flavanones	[63]
Flavone synthase I	1.14.20.5	FNS I	Oxidoreductases	Dioxygenases	Directs 2,3-desaturation of flavanones	[64]
Flavone synthase II	1.14.13.-	FNS II	Oxidoreductases	CYP	Direct conversion of flavanones to flavones	[65]
Flavanone 3 β-hydroxylase	1.14.11.9	F3H	Oxidoreductases	CYP	Catalyzes the 3-beta-hydroxylation of 2S-flavanones to 2R,3R-dihydroflavonols	[66]
Flavanol synthase	1.14.11.23	FLS	Oxidoreductases	Dioxygenases	Formation of flavonols from dihydroflavonols	[67]
Flavonoid 3′-hydroxylase	1.14.14.82	F3′H	Oxidoreductases	CYP	Catalyzes the 3-beta-hydroxylation of 2S-flavanones to 2R,3R-dihydroflavonol	[68]
Flavonoid 3′,5′-hydroxylase	1.14.14.81	F3′5′H	Oxidoreductases	CYP	Catalyzes the conversion of flavones, flavanones, dihydroflavonols, and flavonols to 3′,4′,5′-hydroxylated derivatives	[69]
Leucoanthocyanidin reductase	1.17.1.3	LAR	Oxidoreductases	NADPH reductases	Synthesis of catechin from 3,4-cis-leucocyanidin	[70]
Methyltransferase	2.1.1.-	MT	Transferases	Methyl formers	Transfer of a methyl group from the methyl donor S-adenosyl-l-methionine to substrate	[71]
Anthocyanidin synthase	1.14.20.4	ANS	Oxidoreductases	Dioxygenases	Oxidation of leucoanthocyanidins into anthocyanidins	[72]
Anthocyanidin reductase	1.3.1.77	ANR	Oxidoreductases	NADPH reductases	Catalyzes the double reduction of anthocyanidins, producing a mixture of 2S, 3S and 2S,3R-flavan-3-ols	[73]
Acyltransferase	2.3.1.-	AT	Transferases	Acyl formers	Transfers thioester-activated acyl substrates to a hydroxyl or amine acceptor to form an ester or amide bond	[74]

## 4. Metabolic Engineering of Flavonoid Pathways

Currently, metabolic engineering of flavonoid pathways has gained much attention to enhance the accumulation of specific flavonoid compounds. Several microorganisms, including bacteria and yeasts, and various plant species have been successfully employed as model organisms to engineer flavonoid biosynthetic pathways.

### 4.1. Metabolic Engineering in Plants

Though plant metabolic engineering has been around for the past 50 years, the use of advanced technologies and repeated efforts in recent years have made it possible to achieve increased production of flavonoids by engineering different pathways in plants. The aim of plant engineering is to (1) identify factors involved in flavonoid biosynthesis, (2) enhance pigment production for ornamental purposes, (3) increase plant tolerance to abiotic and biotic factors, and (4) improve flavonoid accumulation [11,75]. While discussing strategies regarding metabolic engineering in plants, it becomes crucial to address the challenges encountering these pathways. The introduction of foreign genes in plants may lead to structural changes. The first step to achieve enhanced flavonoid production is accomplished via the overexpression of key enzymes that actively take part in the biosynthetic pathways [33,70]. Plant transformation to upregulate the enzymes to enhance their metabolic activities is the first obstacle that must be overcome [76]. Secondly, the regulation of the transcriptional and translational mechanism using transcriptional factors is crucial for the successful engineering of metabolic pathways [77]. Thirdly, the reaction may experience an increased metabolic flux (rate of turnover of a metabolite though a reaction system), which must be overcome via careful regulation of the suppression and expression mechanisms of targeted genes to attain desired products. Lastly, the final hurdle during plant engineering is the storage of secondary metabolites because overproduction of the desired product may lead to toxicity due to over-accumulation [78]. This issue must be resolved either by increasing the plant’s storage capacity or through the removal of the accumulated product.

Many plant metabolism-related genes and transcription factors are involved in flavonoid production [79]. Manipulation and regulation of the involved genes and transcription factors could eliminate the rate-limiting steps in numerous plant species, thereby presenting an exceptional opportunity to reveal flavonoid biosynthesis regulatory control. One of the most important families of transcriptional factors involved in the regulation process includes: myeloblastosis (MYBs), which are present both in plants and animals [80,81]. These transcriptional factors were first identified in model plants such as *Arabidopsis*, *Antirrhinum*, *Petunia*, and *Zea*. The overexpression of MYB transcriptional factors has resulted in increased production and enhanced yield with anthocyanin accumulation [82,83]. *Agrobacterium*-mediated genetic transformation (*Ab*-MGT) using MYB transcription factors from *Arabidopsis thaliana* L. [84] into Hop plants resulted in increased flavonoid production. Similarly, overexpression of the MdMyb10 gene resulted in the increased production of flavonoid compounds in the apple plant [85]. The use of MYB transcriptional factors for the enhanced production of flavonoids and to increase the synthesis of pigments to produce colored plants for economic purposes is not new and has been reported previously in many studies [86,87]. Various colors present in plants are due to the presence of anthocyanin pigments, and the flavonoid biosynthesis pathways that lead to the production of these pigments can serve as a potential targets to increase pigment levels in plants [88]. Many studies have successfully reported the enhanced production of anthocyanin by altering the genes involved in flavonoid biosynthetic pathways, including *DFR*, *F3′5′H*, *F3′H*, and *CHS. Agrobacterium*-mediated overexpression of various gene targets in different plant species is shown in Table 2 [89,90]. Moreover, He et al. [91] has reported enhanced production of anthocyanins in dark tobacco varieties, achieved through metabolic engineering. Furthermore, *Nicotiana* spp. and legumes have served as potential targets for metabolic engineering to produce plants with enhanced proanthocyanidin and anthocyanin concentrations [92,93]. The use of tissue-specific promoters and transcriptional factors for the production of *Zea mays* and *Saussurea involucrata* rich in anthocyanin pigments has been described as well [94,95]. Some potential gene targets for the enhanced production of flavonoids have been mentioned in Table 3 for various plant species. Recent advancements in technologies have also started to undermine the challenges in enhanced flavonoid production. One widely used strategy is reverse genetic engineering (RGE). This approach employs the use of gene silencing techniques, point mutations, gene insertions, and knock-out tactics for the engineering of plant flavonoid biosynthesis pathways. The goal is eventually to produce plants capable of producing flavonoids with enhanced quantity [96]. RGE approaches, such as RNA interference silencing, have been used for engineering *F3′5′H*, F3′H, CHS, CHI, and *DFR* genes in various plants, including tomato, maize, rice, tobacco, and beans, for improved flavonoid production [11,33,36]. Similarly, the use of plant tissue culture technology to produce engineered plants with enhanced flavonoid production has also been employed [97]. No matter which technique is employed, metabolic engineering is a reliable and efficient way of producing plants with high flavonoid concentrations and may become an indispensable tool for large-scale flavonoid production in the years to come.

### 4.2. Metabolic Engineering in Microbes

Being an economic commodity, the net worth for the global markets of flavonoids is currently estimated to be >US $200 million per year [125]. The existing methods for flavonoid production depend primarily on plant extraction and chemical synthesis [24]. However, in the plant extraction procedure, minimal yields and complex downstream purification procedures are the main hurdles that need to be overcome [126]. Other challenges faced in the extraction of flavonoid compounds from plants that limit their large-scale production are (a) hard-to-culture plant species, (b) the breeding period, (c) climate, and (d) slow growth rates of plants [5,20]. In chemical synthesis methods, the need for noxious chemicals and stringent reaction conditions restricts the de novo synthesis of such secondary metabolic compounds. For the pharmaceutical industry, preparing pure flavonoids is a major hurdle [20]. Hence, flavonoid biosynthesis by microbes has received attention as an appealing alternative due to the utilization of environmentally friendly feedstocks, low energy demands, and reduced waste emissions via the heterologous reconstruction of plant biosynthetic gene clusters into the microbes [20,125,127,128]. Different flavonoid pathways were reconstituted in several microbial species in the past few decades (Figure 3), including *Saccharomyces cerevisiae* and *Escherichia coli*, leading to the synthesis of various flavonoids, including baicalein, silybin, resveratrol, eriodyctiol, naringenin, anthocyanins, pinocembrin, kaempferol, scutellarein, quercetin, isosilybin, and so forth [129,130].

Metabolic engineering modifies the microorganism’s function through the insertion of non-native DNA sequences, resulting in the production of enzymes that can synthesize the target compound. With the help of the precursors already present in the host microorganism, this bioengineered enzymatic system mimics and reproduces the flavonoid biosynthetic pathway in plants [131,132]. Metabolic engineering of microorganisms for flavonoid synthesis involves host strain selection, optimization of conditions, target identification for genetic manipulations, and understanding the enzymes associated with plant biosynthetic pathways. Generally, the metabolic engineering in microorganisms for flavonoid biosynthesis involves the following steps: recombinant pathway engineering and bioprospecting; selection, cloning, or heterologous gene construction; choice of vector and production host, and heterologous gene transformation into the host; optimization of the expression, folding, and activity of phytoproteins via protein engineering in the microbial hosts; improvement of the strain through redistribution of the carbon flux, reductions in toxicity, transporter engineering, regulatory restriction removal, enzyme compartmentalization or co-localization, and pathway balancing [126,133,134,135]. Common metabolic strategies for enhancing the yields of flavonoids include the (a) removal of feedback inhibition points, (b) enhanced availability of precursors, (c) blocking of competitive pathways, (d) strain-dependent pathway balancing, and (e) co-factor recycling and biosynthesis [136]. These strategies are reviewed elsewhere in greater detail [137,138].

Metabolic engineering or pathway reconstruction of microbes utilizing novel enzymes and synthetic biology tools results in the fabrication of a range of compounds (such as stilbenes, anthocyanins, flavonoids, and others) and their natural and synthetic derivatives [126]. Optimization of microbial strains and their metabolic pathways, the regulation of these pathways, and tolerance engineering has led to the creation of microbial cell factories by introducing plant metabolic pathways into microbes to produce the desired compounds on a commercial scale in an economical and eco-friendly manner [126]. Moreover, studies have shown that microbial fermentation is easily scalable from the laboratory to commercial scales of production. Furthermore, the simplicity with which these microorganisms can be genetically modified, as well as the accessibility of molecular techniques (for example, heterologous polyphenol pathway gene expression, homologous polyphenol pathway gene manipulation, or genome editing), makes it possible to construct microbial cell factories for the production of almost any natural or artificial metabolite possible [10].

Microbial production has shown immense promise in the synthesis of natural compounds derived from various plant species because of the fast growth of microbes, easy cultivation, and advanced microbial engineering techniques, such as versatile genetic manipulation and easily accessible bioinformatics tools, allowing the microbial processing of natural products in a simple, manageable, and cost-effective manner [31,139,140]. Furthermore, microbial biosynthesis is much more eco-friendly as compared with chemical synthesis because it can reduce the need for harmful organic solvents or other required chemicals for product purification. Thus, the microbial cultures can produce pure products in contrast to chemical synthesis, allowing for easier and “greener” purification techniques [10,31]. In conjunction with the advancements in DNA synthesis and genome sequencing, enzymology and structural biology and the simulation of metabolic networks and metabolic engineering play a significant and crucial part in the production of on-demand metabolite possessing for potent biochemical activities and are of industrial relevance from microbes [140]. Nevertheless, the practical reconstitution of biosynthetic pathways of plants in microorganisms, as well as the implementation of microbial synthesis for the industrial production of essential compounds, remains challenging [31]. Several studies reported the biosynthesis of various flavonoids from different microbial species (Table 4).

The hydroxylation of flavonoids is the most integral post-modification process used for a wide variety of flavonoids to be biosynthesized. To produce heterogeneity in flavonoid structural chemistry and to create unique therapeutically or nutraceutically active compounds, flavonoids are hydroxylated at the C3′ position via the action of several hydroxylase enzymes, such as flavanone 3′-hydroxylase. This process of hydroxylation occurs predominantly at carbons (C3, C6) and at the B ring, involves cytochrome P450 reductases that are present on the ER membrane, and aids in the transfer of electrons to receptors [141]. The hydroxylation of the carbon atoms in the flavonoid compounds increases their solubility and boosts their metabolic stability, thus significantly improving their biological properties [130,142]. To date, hydroxylated flavonoid production remains a major challenge because P450-related enzymes of plants are not effectively expressed in microorganisms [130]. Moreover, the hydroxylation of flavonoids has been studied less, as compared to methylation and glycosylation processes [143]. Lv et al. [130] used a modular approach for the construction, characterization, and optimization of flavonoid pathways in *Yarrowia lipolytica* for the synthesis of flavonoids and hydroxylated flavonoids. Cytochrome P450 reductase (CPR) and chalcone synthase (CHS) are known to impede the production of hydroxylated flavonoids. After the removal of precursor pathway limitations and further optimization, their engineered strain produced 15.8-fold more naringenin (252.4 mg/L), 6.9-fold more eriodictyol (134.2 mg/L), and 8.8-fold more taxifolin (110.5 mg/L) from glucose in the flasks. Recent advancements in the hydroxylation of isoflavones by microbes have been reviewed elsewhere [143]. In recent years, more progress has been made toward increasing eriodictyol production and introducing metabolic pathways in various microbes. Gao et al. [144] reported the highest eriodictyol titer (3.3 g/L) to date by identifying and optimizing efficient *Sm*CPR and *Sm*F3′H from *Silybum marianum* in *S. cerevisiae*, demonstrating that *S. cerevisiae* is an attractive platform for P450 enzyme expression involved in flavonoid biosynthesis. In another study, *Corynebacterium glutamicum* (a GRAS strain) was used as a host for the first time to directly produce eriodictyol (14.10 mg/L) from tyrosine by introducing *matC* and *matB* genes from *Rhizobium trifolii*, as well as hpaBC genes expressing 4-hydroxyphenylacetate 3-hydroxylase from *Escherichia coli* [145].

Recent research has shown that flavonoid methylation could expand their potential as pharmaceutical agents, contributing to novel applications. Flavonoid methylation on their free carbon atoms or hydroxyl groups significantly improves their metabolic stability and facilitates transport through the membrane, resulting in better absorption, improved oral bioavailability, and solubility [146]. Koirala et al. [147] designed a bioconversion system using genetically modified *E. coli* as a host harboring *E. coli* K12-derived metK encoding S-adenosylmethionine (SAM) synthase enzyme and a *Streptomyces avermitilis*-originated O-methyltransferase (SaOMT2) for the methylation of genistein and daidzein. An increase in the yield of 4′-*O*-methyl genistein (46.81 mg/L) and 4′-*O*-methyl daidzein (102.88 mg/L) was successfully achieved via whole-cell region-specific biotransformation.

Microbial glycosylation of flavonoids is an efficient strategy since it modulates many of their pharmacokinetic parameters, such as improving their water solubility, bioavailability, and stability, therefore enhancing the potential application of bioactive compounds on a large scale [148,149,150,151]. Glycosylation is an important modification strategy that has been used for a long time and has resulted in the discovery of many novel metabolites that are not found naturally. A metabolic engineering approach for quercetin modification through glycosylation in *E. coli* involves overexpression of the corresponding glycosyltransferases (GTs) in the microbial host and metabolic engineering to increase nucleoside diphosphate-sugar (NDP-sugar) [152]. Glycosyltransferase family 1 (GTF1) enzymes are considered ideal biocatalysts for flavonoid glycodiversification, despite their relatively low yields [148]. Several studies on the microbial glycosylation of flavonoids have recently been reported, such as that of Pei et al. [153], which described the enhanced production of astragalin and kaempferol from flavanone (naringenin) via metabolic engineering in an *E. coli* strain. By optimizing the fermentation conditions, the maximum production of kaempferol (1184.2 mg/L) has been reached, representing the highest kaempferol yield recorded to date from naringenin. To produce astragalin, glycosyltransferase and a pathway for UDP-glucose synthesis have been inserted into the recombinant *E. coli* strain, resulting in the maximum synthesis of astragalin (1738.5 mg/L) with no accumulation of kaempferol. Similarly, Ruprecht et al. [151] introduced a novel metabolic engineering strategy via whole-cell biotransformations to enhance the rate of rhamnosylation in *E. coli* UHH_CR5-A by deleting the genes for UTP-glucose-1-phosphateuridyltransferase (galU) and phosphoglucomutase (PGM) and simultaneously overexpressing the genes for dTDP-rhamnose synthesis (rmlABCD) and glucan 1,4-alpha-maltohexaosidase enzymes, for enhanced degradation of maltodextrin, next to glycosyltransferase (CGtfC). These modifications in *E. coli* UHH_CR5-A led to a 3.2-fold increase in the production of hesperetin rhamnosides compared to that in an *E. coli* MG1655 strain expressing GtfC in 24 h batch-type fermentations. Moreover, after a duration of 48 h, *E. coli* UHH-CR_5-A could produce a final product volume of 2.4 g/L of hesperetin-3′-orhamnoside. Dou et al. [150] used *Isaria fumosorosea*, which is an entomopathogenic filamentous fungus for the biotransformation of a group of natural phenolic products into their glycosides, comprising anthraquinone and flavonoids, hence resulting in the formation of six new flavonoid (4-*O*-methyl) glucopyranosides.

*E. coli* is always the host of choice for flavonoid production because of several benefits: (a) it is a well-studied and widely recognized biological workhorse, (b) it requires a readily available rich medium, and (c) it has a short doubling-time, which reduces the time constraint for product extraction [5]. The production of flavonoids in *E. coli* strains has been reported in several studies, such as that of Shrestha et al. [154], which developed a synthetic vector system by cloning two essential genes: cyanidin 3-*O*-glucosyltransferase (At3GT) and anthocyanidin synthase (PhANS) from *Arabidopsis thaliana* and *Petunia hybrida* under Ptrc, PT7, and PlacUV5 promoters to produce a combination system for the synthesis of cyanidin 3-*O*-glucoside (C3G) in *E. coli* by driving the metabolic flux to UDP-d-glucose. One of the metabolically engineered strains containing PhANS and At3GT under the Ptrc promoter and biosynthetic genes for UDP-d-glucose under the PT7 promoter resulted in the enhanced production of C3G (~439 mg/L) within 36 h of incubation. Similarly, Li et al. [155] created an artificial pathway, for the first time, for the production of flavones (scutellarein and baicalein) by reconstituting plant pathway genes for flavonoid biosynthesis from five different species in an *E. coli* cell factory: 4-couramate-coenzyme A ligase (4CL) derived from *Petroselinum crispum*, PAL derived from *Rhodotorula toruloides*, CHI derived from *Medicago sativa*, CHS derived from *Petunia hybrida*, and an oxidoreductase flavone synthase I (FNSI) derived from *P. crispum*. Tyrosine and phenylalanine were used as precursors for the production of the intermediates, such as apigenin and chrysin. Recently, Kim [128] successfully managed to biosynthesize genistein from either naringenin or *p*-coumaric acid utilizing *E. coli* BL21 (DE3) as a biotransformation host. Four genes, 4-coumarate-CoA ligase (Os4CL), chitin synthase (PeCHS), cytochrome P450 reductase (OsCPR), and isoflavone synthase (RcIFS), were used in the biosynthesis of genistein. Using *p*-coumaric acid and naringenin, genistein was expressed at up to 18.6 mg/l and 35 mg/l, respectively, under optimized conditions of the culture. In the study by Whitaker et al. [156], the authors engineered *E. coli* to convert methanol into flavanone naringenin by incorporating pathway enzymes of ribulose monophosphate and methanol dehydrogenases from *Bacillus methanolicus* and *Bacillus stearothermophilus*.

Apart from *E. coli*, flavonoid production has been reported in other bacterial strains as well. Zha et al. [139] successfully produced anthocyanidin 3-*O*-glucoside (C3G) from a relatively inexpensive flavonoid precursor, catechin by coexpressing 3-*O*-glucosyltransferase (3GT) and anthocyanidin synthase (ANS) from *Arabidopsis thaliana* and *Petunia hybrida* in *Corynebacterium glutamicum*. After optimization, C3G production (~40 mg/L) was observed, reflecting a 100-fold increase in the titer relative to production in an un-optimized, un-engineered starting strain. In a similar study, Solopova et al. [157] demonstrated that engineered strains of *Lactococcus lactis* could rapidly turn inexpensive flavan-3-ol-rich sources (for example, green tea infusion) into a variety of particularly useful plant-derived bioactive anthocyanins.

*Saccharomyces cerevisiae* is an appealing microbe that can be used as a host for de novo flavonoid production, owing to its capability for posttranslational modifications (PTMs) of eukaryotic proteins, resulting in improved expression of phytoproteins. Unlike *E. coli*, yeast has a greater capability to rapidly express type II P450 hydroxylase enzymes, several of which are key catalysts in flavonoid biosynthetic pathways [5,20,136]. Jiao et al. [158] used a *S. cerevisiae* (strain Y-01) for the synthesis of naringenin and indicated that its production is not affected by CHI or 4CL expression, while a positive correlation was seen between CHS expression and naringenin production. Similarly, Mar et al. [18] used the gene source screening approach as a method for determining the appropriate source of genes for enzymes, such as CHS and 4CL. The *Vitis vinifera* CHS gene and the *Medicago truncatula* 4CL gene have been expressed for the first time in *S. cerevisiae*, and this combination resulted in the maximum production of naringenin, which was 28 times greater than that in the reference strain. Lyu et al. [159] constructed a high-yield cell factory for the highest kaempferol production (86 mg/L) in *S. cerevisiae* (strain YL-4) through different approaches, such as the removal of the phenyl ethanol biosynthetic branch, screening of genes, supplementation of precursor phosphoenolpyruvate (PEP)/D-erythrose 4-phosphate (E4P), optimization of the core flavonoid synthetic pathway, and mitochondrial engineering of flavonol synthase (FLS) and flavanone 3-hydroxylase (F3H). In recent research, Borja et al. [2] employed a combination of metabolic engineering and system biology approaches to develop a strain of *S. cerevisiae* capable of producing *p*-coumaric acid (242 mg/L) from the monosaccharide xylose.

De novo flavonoid production has recently been reported in other yeasts and fungi as well. *Yarrowia lipolytica*, a non-conventional yeast, is a superior and promising host for the manufacturing of natural plant-based products because of its innovative features relative to conventional hosts [24]. Lv et al. [130] engineered *Y. lipolytica* to produce naringenin (252.4 mg/L), eriodictyol (134.2 mg/L), and taxifolin (110.5 mg/L) from glucose in shake flasks. In another study, *Y. lipolytica* was designed to biosynthesize 12.4 g/L of resveratrol in a controlled fed-batch bioreactor [160]. Similarly, Bu et al. [161] has reported the biosynthesis of anthocyanin for the first time using fungi (*A. sydowii* H-1 strain). For this purpose, 31 candidate transcripts were recognized, in which cinnamate-4-hydroxylase gene (C4H) and chalcone synthase gene (CHS), identified as the main genes in the biosynthesis of anthocyanin, were found only in the H-1 strain, which suggested that these two genes might have contributed to the production of anthocyanin in the H-1 strain.

Due to advancements in synthetic and system biology, the microbial biosynthesis of flavonoids and their precursors has increased to more than 1 g/L. Chen et al. [162] employed *S. cerevisiae* to engineer the recycling and supply of three cofactors (S-adenosyl-L-methionine, NADPH, and FADH2) for high-level ferulic acid (3.8 g/L) and caffeic acid (5.5 g/L) production, demonstrating the importance of cofactors in natural product biosynthesis. In another study, the engineering of the aromatic amino acid biosynthesis pathway in *S. cerevisiae* resulted in an increased titer (12.5 g/L) of *p*-coumaric acid [163]. In a 5 L bioreactor, *S. cerevisiae* biosynthesized 1.21 g/L of (2S)-naringenin from *p*-coumaric acid. This was accomplished by selecting strong promoters for the optimization of the biosynthesis pathway [164]. Similarly, Zhang et al. [165] increased the (2S)-naringenin titer (1129.44 mg/L) in *S. cerevisiae* by utilizing fatty acid β-oxidation in the *S. cerevisiae* peroxisomes for the enhancement of cytoplasmic acetyl-CoA levels. Significant improvements have also been made in the biosynthesis of various flavonoids in microbes, especially the post-translationally modified flavonoids. Li et al. [166] metabolically engineered *S. cerevisiae* by integrating flavonoid 3′-hydroxylase and flavonoid 3-hydroxylase, as well as TDH1p and INO1p promoters, for enhanced dihydromyricetin production (709.6 mg/L) in a 5 L bioreactor from naringenin (2.5 g/L). *S. cerevisiae* was engineered to de novo produce two basic flavan-3-ols, catechin and afzelechin, from an already engineered *S. cerevisiae* strain, E32, which could produce naringenin using glucose as a substrate. The incorporation of F3′H, F3H, DFR, and LAR led to de novo catechin and afzelechin production. Increased NADPH supply and promoter optimization further enhanced their production. After fermentation in a 5 L bioreactor for 90 h, the optimal strains produced a total of 321.3 mg/L catechins and 500.5 mg/L afzelechin, respectively [167]. Lyu et al. [69] isolated UDP-rhamnose synthase and three novel 3-*O*-glycosyltransferases from a traditional Chinese herb, i.e., *Epimedium koreanum* Nakai, and expressed them in *S. cerevisiae* to assess their potential for flavonol glycosylation. Kaempferol was utilized as a substrate, and the engineered strain was able to bioproduce kaempferol-3-*O*-galactoside (130.9 mg/L), kaempferol-3-*O*-rhamnoside (5.5 mg/L), and kaempferol-3-*O*-glucoside (13.5 mg/L) after 144 h using a fed-batch approach. A recent study established the biosynthesis of two flavone 8-*C*-glycosides (vitexin and orientin) from luteolin and apigenin, respectively, in *E. coli* by isolating *C*-glucosyltransferase from *Trollius chinensis* and coupled it with sucrose synthases to increase vitexin (5524.1 mg/L) and orientin (2324.4 mg/L) yields [168].

Increasing the number of intracellular precursors is the conventional method for enhancing flavonoid production. PEP precursor intracellular transport is mediated by the phosphotransferase system (PTS) [163]. Inhibiting the PTS system is one way of increasing the proportion of PEP that attempts to enter the shikimate pathway. Due to the elimination of the PTS, a low glucose-consumption rate may be found. This dilemma can be resolved by substituting PTS with glucose transport systems in which PEP is not consumed. The PEP supply can also be increased through the overexpression of PEP synthase (*PpsA*). Inactivation of the PTS system increases L-phenylalanine production (72.9 g/L) in *E. coli* (strain Xllp01), as reported by Liu et al. [169]. In addition to PEP, E4P is an additional precursor originating from the pentose phosphate pathway. The overexpression of the transketolase and transaldolase enzymes, which are encoded by the *talB* and *tktA* genes, respectively, is a widely accepted method for increasing the intracellular concentration of E4P [170]. Fordjour et al. [171] described enhanced 4-dihydroxyphenyl-L-alanine (L-DOPA) production from D-glucose by selectively repressing and activating particular genes in *E. coli* BL21 (DE3), resulting in the accumulation of both precursors (E4P and PEP).

Despite the recent advancements in microbial flavonoid biosynthetic processes, several problems remain to be addressed to enhance productivity and increase the market viability of microbial production. The limitations associated with this method include improper gene balancing linked to the microbial flavonoid biosynthetic pathways, poor expression, and transporter engineering for the effective exudation of the metabolites into the extracellular medium [140].

Various other factors that hamper large-scale flavonoid production utilizing recombinant microorganisms include a lower malonyl CoA intracellular concentration, a precursor metabolite for flavonoid synthesis. Progress and selective approaches for enhancing malonyl-CoA production in *S. cerevisiae* have been described in detail by Li et al. [172]. Several strategies exist that can be used for enhancing malonyl-CoA production. In a recent study, Zhou et al. [173] demonstrated that it was possible to produce naringenin by controlling intracellular levels of malonyl-CoA using a three-layered dynamic regulation network in *E. *coli**, yielding naringenin at a concentration of 523.7 ± 51.8 mg/L with an 8.7-fold improvement. Similarly, another study demonstrated that a metabolic engineering approach enhanced malonate synthesis by targeting malonyl-CoA. The process involved improving the catalytic activity of the 3-hydroxyisobutyryl-CoA hydrolase enzyme by inducing targeted mutations in *the Ehd3* gene in *S. cerevisiae* (strain LMA-16). Results indicated malonate production of 187.25 mg/L in the flask and a total of 1.62 g/L in fed-batch fermentation [174].

Other factors include a lack of the supply of aromatic amino acids, including phenylalanine and tyrosine (the two major precursors of phenylpropanoic acids), the cellular metabolism interconnectivity leading to unstable phenotypes, the weak expression of some metabolic pathway enzymes, and the low solubility and instability of certain flavonoids. With the advent of systems metabolic engineering, a blend of synthetic biology, systems biology, and systems-level evolutionary engineering, new perceptions and insights into the improvement of production strains and processes have emerged [126,175].

To optimize flavonoid production by microbial cell factories for commercial applications, several efforts were made to optimize the incorporated pathways, such as the selection of promoters, ribosome-binding sites, and gene orthologues, product relocation, regulation, and the improvement of the co-substrate supply. The resulting strains, however, do not meet the requirements set for industrial-scale applications. Considering the significance of pathway enzymes in successful bioproduction and the fact that extensive knowledge of the majority of these enzymes is still being explored, it is crucial to glean new insights from these enzymes and to suggest novel strategies for optimizing metabolic pathways and improving production titers [141].

A better understanding of genomes and their regulatory role in metabolic pathways of microbes has facilitated genome-scale engineering of pathways and systems metabolic engineering, which utilize data-driven methods, such as the *in silico* and omics-based prediction of pathways and techniques for gene selection to design, modulate, and optimize metabolic pathways and protein function evolution. Approaches based on *in silico*-aided metabolic engineering have accelerated the development of microbial strains for the commercial production of biochemicals and amino acids, which is due to the improved efficiency and capability of engineered microorganisms for scale-up bioprocessing and fermentation. The rapid advancement of next-generation omics technologies and synthetic biology techniques, such as CRISPR interference (a gene editing technology that enhances gene expression) will improve the utilization of substrates and the development of hyper-producing strains through genome-wide evaluations and high-throughput screening of strains [176].

## 5. Microbial Co-Culture Strategy for Flavonoid Biosynthesis

Co-culture engineering offers a robust and efficient toolbox for complex compound biosynthesis with long biosynthetic pathways. The engineering of microbial consortia for the expression of complex biosynthetic pathways of flavonoids is a promising approach for flavonoid production and has been extensively researched [20,182]. However, existing microbial biosynthesis strategies depend primarily on the use of a single strain, generated via metabolic engineering, to accommodate the complicated and long flavonoid pathways, which is an efficient way to biosynthesize valuable compounds, but suffers from many drawbacks, such as severe metabolic imbalances, failure to achieve optimum conditions for all pathway-specific enzymes to function, increased metabolic burden because of the reconstruction or heterologous expression of complicated metabolic pathways, and unwanted by-product accumulation, making it difficult to increase the yields of the final product [128,182,183]. Recently, co-culture systems utilizing two or more microbial strains were developed to resolve these concerns and improve final titers [128] (Figure 4). Compared to the conventional monoculture strategy, modular co-culture engineering exploits the power of multiple strains of microbes to reconstruct the target biosynthetic pathway. As a result, it significantly reduces the biosynthesis labor of each microbial strain, as well as the metabolic burden associated with it, and it also restricts the formation of undesired by-products [183,184]. These benefits are exceptional for the biological synthesis of diverse natural plant products, involving intricate lengthy pathways, and requiring extensive yet delicate engineering efforts [182]. Microbial co-cultures provide numerous advantages over monocultures in terms of removing pathway bottlenecks and enhancing metabolic productivity. These advantages include metabolic labor division, the sharing of gene expression burdens, and the ability to allow cross-feeding of metabolites to help improve pathway performance and metabolic robustness. A complete biosynthetic pathway is modularized using the most extensively used co-culture approach, with each module tailored to achieve the maximum production volume and intermediate metabolite yield. In co-culture, metabolic intermediates or precursors must be available and transported to the downstream strains [125]. Moreover, cellular specialization and compartmentalization allow microbial consortia to cope with major environmental fluctuations and to perform complicated tasks that individual members would be unable to carry out. Thus, microbial co-cultures can improve productivity and precision by presenting a simple way of optimizing each submodule by dividing the metabolic burden and reducing the strain on members of the consortia present in the culture [125].

Microbial co-cultures have been used in recent studies to produce complex flavonoids (Table 5). The evolving modular co-culture engineering method was adapted by Wang et al. [182] to rationally design, develop, and optimize an efficient *E. coli* co-culture for sakuranetin production from glucose. The upstream *E. coli* strain was designed for the intermediate *p*-coumaric acid synthesis pathway, while *p*-coumaric acid was converted to sakuranetin by the downstream strain. Via stepwise co-culture system optimization, sakuranetin (29.7 mg/L) was produced from 5 g/L glucose over a duration of 48 h, which was substantially elevated in comparison to that with the typical monoculture-based synthesis approach. Similarly, Akdemir et al. [185] co-cultured recombinant strains of *E. coli* to produce pyranoanthocyanins with increased yields and titers. First, they engineered 4-vinylcatechol (4VC)- and 4-vinylphenol (4VP)-producing modules to achieve this task and then co-cultured both strains with recombinant cells producing C3G to obtain pyranocyanidin-3-*O*-glucoside-phenol (C3G with 4VP adduct) and pyranocyanidin-3-*O*-glucoside-catechol (C3G with 4VC adduct). Thuan et al. [183] presented a co-culture system based on genetically engineered *E. coli* for the de novo biosynthesis of apigetrin. The culture system consisted of an upstream module to synthesize apigenin from p-coumaric acid that included chalcone isomerase, chalcone synthase, 4-coumarate:CoA ligase, and favone synthase I. Meanwhile, the downstream system is comprised of a metabolic module to increase UDP-glucose production and glycosyltransferase expression to convert the apigenin to apigetrin. A yield of apigetrin (16.6 mg/L) was obtained with the help of this co-culture approach. Recently, Du et al. [20] compared the pathway efficacy between the monoculture and co-culture of *S. cerevisiae* for flavonoid production. The delphinidin titer produced via co-culture was considerably higher than that with monoculture of the strain sDPD2, which harbored the entire pathway. Recently, eriodictyol was produced from D-glucose using a co-culture system based on *E. coli*. The first *E. coli* strain possessed genes for *p*-coumaric acid synthesis from D-glucose, while the genes for eriodictyol biosynthesis from *p*-coumaric acid were contained in the second strain. The co-culture produced a higher eriodictyol yield of 51.5 mg/L than that with monoculture, which achieved 21.3 mg/L [186].

## 6. Conclusions

Flavonoids encompass a large group of polyphenolic secondary metabolites in plants and have gained considerable attention over the past few decades. In recent years, several studies have not only demonstrated the beneficial aspects of flavonoids in terms of the treatment of numerous human conditions but also determined their potential industrial applications as well. The global market for naturally derived flavonoids is expected to reach $1.5 billion by the end of 2025. Conventional methods for the extraction of flavonoids have several limitations, and they cannot meet this rising demand of the global market. Thus, to overcome the limitations of traditional methods and for the scalable production of flavonoids, various methodologies have evolved to enhance the production of flavonoids both in plants and microbes through the metabolic engineering of different flavonoid pathways. Regulation of gene expression at different levels using several modulators, the reconstruction of biosynthetic pathways in microbes, and the efficient use of co-culture strategies have paved the way for large-scale success in flavonoid production. Similarly, the emergence of advanced microbial systems and synthetic biology technologies have brought about a revolution in the metabolic engineering of phytochemicals. In the future, employing novel gene-editing tools, such as CRISPR-Cas9, along with synthetic biology approaches will be widely used for the tailor-made production of phytochemicals in bioengineered microbes, particularly flavonoids. Researchers are also targeting their efforts toward discovering new microbial species that can produce multiple favorable compounds efficiently. Even though several limitations need to be addressed in terms of improving costs and developing economically viable processes for the large-scale production of flavonoids, the list of commercially available flavonoids is expected to grow rapidly in the coming years due to the advances in technologies, thus further augmenting their role in therapeutics and phytomedicine.

## Figures and Tables

**Figure 1 metabolites-13-00124-f001:**
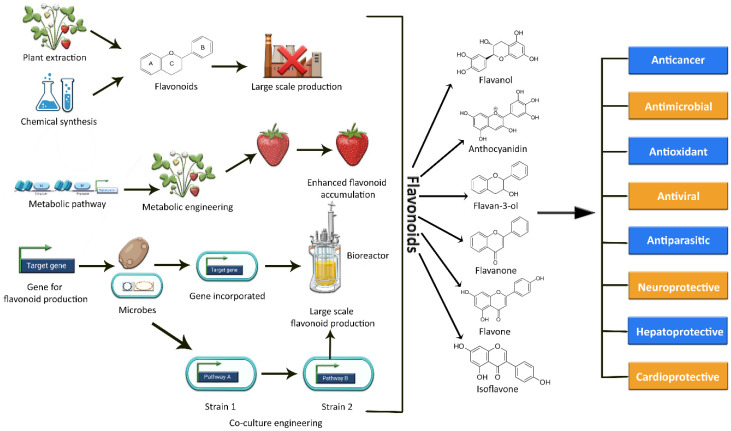
Graphical representation of the various methods used to synthesize a variety of flavonoids, as well as their potential biological activities.

**Figure 2 metabolites-13-00124-f002:**
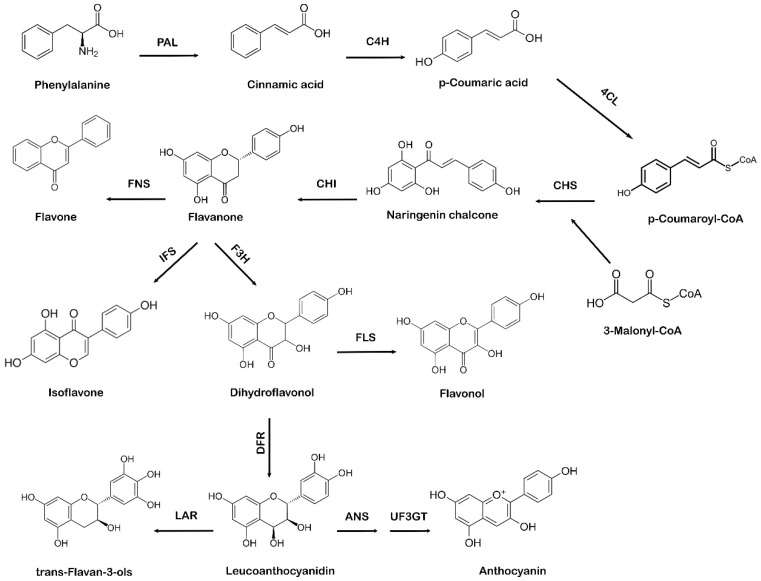
Systematic representation of flavonoid biosynthesis. PAL: phenylalanine ammonia-lyase, C4H: cinnamate-4-hydroxylase, 4CL: 4-coumarate:CoA ligase, CHS: chalcone synthase, CHI: chalcone isomerase, IFS: isoflavone synthase, F3H: flavanone 3 β-hydroxylase, FNS: flavone synthase, FLS: flavonol synthase, DFR: dihydroflavonol 4-reductase, ANS: anthocyanidin synthase, UF3GT: anthocyanidin 3-*O*-glucosyltransferase, LAR: leucoanthocyanidin reductase.

**Figure 3 metabolites-13-00124-f003:**
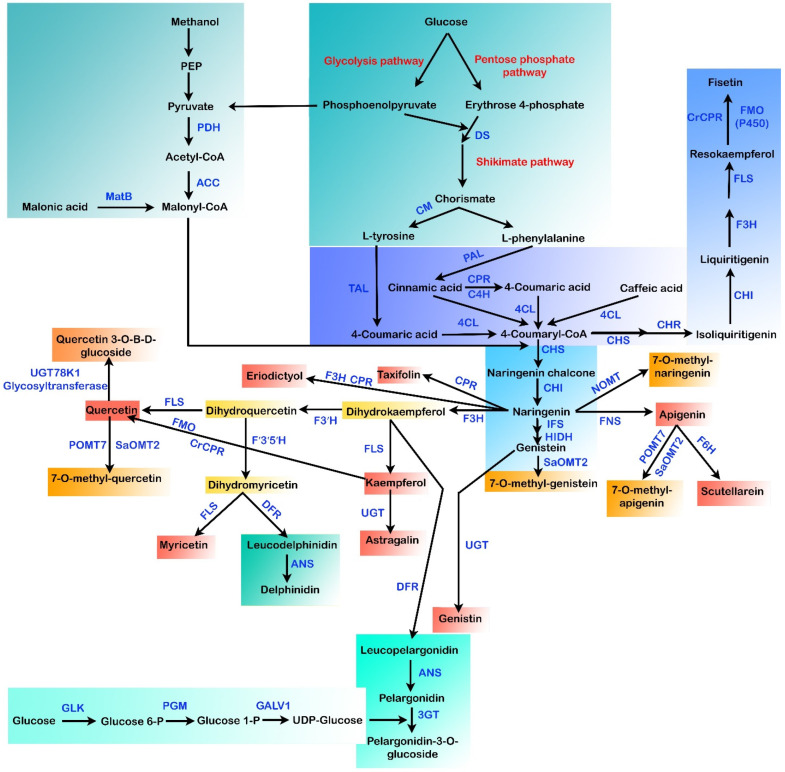
Different biosynthetic pathways for the microbial production of flavonoids. MatB: malonyl-CoA synthetase, DS: 3-deoxy-D-arabinoheptulosonate 7-phosphate (DAHP) synthase, ACC: acetyl-CoA carboxylase, PDH: pyruvate dehydrogenase, CM: chorismate mutase, TAL: tyrosine ammonia-lyase, PAL: phenylalanine ammonia-lyase, CPR: cytochrome P450 reductase, C4H: cinammate-4-hydroxylase, 4CL: 4-coumaroyl-CoA ligase, CHI: chalcone isomerase, CHR: chalcone reductase, CHS: chalcone synthase, IFS: isoflavone synthase, POMT7: apigenin-7-*O*-methyltransferase, HIDH: 2-hydroxyisoflavanone dehydratase hydroxy type, NOMT: naringenin *O*-methyltransferase, SaOMT2: *Streptomyces avermitilis O*-methyltransferase, F3H: flavanone-3-hydroxylase, FNS: flavone synthase, F6H: flavone 6-hydroxylase, FLS: flavonol synthase, FMO: flavin-containing monooxygenase, F3′H: flavonoid-3′-hydroxylase, UGT: uridine diphosphate glycosyltransferase, 3GT: anthocyanidin 3-*O*-glycosyltransferase, ANS: anthocyanidin synthase, F’3′5′H: flavonoid 3′,5′-hydroxylase, GLK: glucokinase, DFR: dihydroflavonol 4-reductase, PGM: phosphoglucomutase, GALU1: UTP-glucose-1-phosphate uridylyltransferase.

**Figure 4 metabolites-13-00124-f004:**
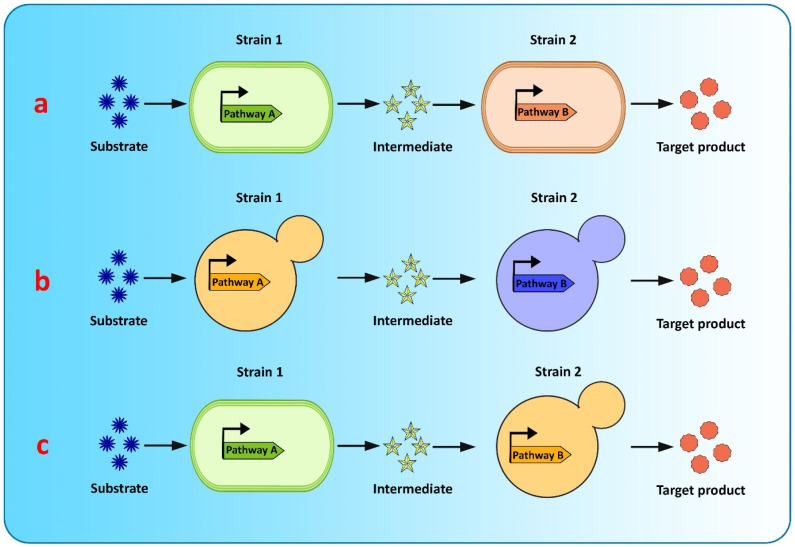
Graphical representation of modular co-culture approach for de novo biosynthesis of flavonoids (**a**) *E. coli*–*E. coli* co-culture (**b**) *S. cerevisiae*–*S. cerevisiae* co-culture (**c**) *E. coli*–*S. cerevisiae* co-culture.

**Table 2 metabolites-13-00124-t002:** Flavonoid production using metabolic engineering approaches in various plant species.

Plant Species	Gene Target	Enhanced Flavonoids Produced	Reference
*Camellia sinensis* L.	*CsAN1*	Anthocyanin	[80]
*Nicotiana benthamiana* and *Lilium*	*ROSEA1* and *DELILA*	Anthocyanin	[81]
*Solanum lycopersicum*	*LeAN2*	Anthocyanin	[98]
*Humulus lupulus* L.	*PAP1/AtMYB75*	Anthocyanin	[99]
*Scutellaria bornmuelleri*	*MYB7* and *FNSП2*	Chrysin, Wogonin, and Baicalein	[100]
*Marchantia polymorpha*	*R2REMB*	Anthocyanin	[101]
*Dracaena cambodiana*	*DcCHI1* or *DcCHI4*	Anthocyanin	[102]
*Arabidopsis thaliana*	*CsCYT75B1*	Anthocyanin	[103]
*Ginkgo bilboa*	*GbF3′H1*	Flavanones	[104]
*Glycyrrhizia uralensis*	*CHS*	Flavanones	[105]
*Malus domestica*	*MdMyb10*	Anthocyanin	[85]
*Nicotiana tabacum*	*AtPAP1*	Anthocyanin	[91]
*Salvia miltiorrhiza*	*SmMYC2*	Anthocyanin	[106]
*Salvia miltiorrhiza*	*SmJMT*	Flavanones	[107]
*Salvia miltiorrhiza*	*SmANS*	Anthocyanin	[108]
*Talinum paniculatum*	*GmCHI*	Flavanones	[109]
*Petunia hybrida*	*Fh3GT1*	Anthocyanin and Flavonol	[110]
*Apium graveolens*	*AgMYB12*	Apigenin	[111]
*Aconitum carmichaelii*	*F3′5′H*	Flavanones	[112]
*Astragalus trigonus*	*chiA*	Apigenin	[113]

**Table 3 metabolites-13-00124-t003:** Potential gene targets for metabolic engineering in plant flavonoid biosynthesis pathways.

Plant Species	Gene Targets	Reference
*Paeonia suffruticosa* Andr	*F3′H*, *F3′5′H*	[114]
*Brassica napus*	*FNS I*, *FNS II*	[115]
*Hordeum vulgare*	*IFS*, *IFR*	[116]
*Allium cepa* L.	*ANS*, *FLS*	[117]
*Artemisia annua* L.	*PCL*, *PAL*	[118]
*Glycine* spp.	*ANS*, *FLS*	[119]
*Ginkgo biloba*	*DFR*, *ANR*	[120]
*Camellia sinensis*	*LAR*, *ANS*	[121]
*Salvia miltiorrhiza*	*ANS*, *ANR*	[32]
*Mangifera indica*	*CHI*, *CHS*, *CHR*	[122]
*Oroxylum indicum*	*CHI*, *CHS*	[123]
*Nicotiana tabacum*	*ANS*, *FLS*	[124]

**Table 4 metabolites-13-00124-t004:** Mono-culture approach for the microbial production of flavonoids.

Substrate	Product	Host Strain	Titer (mg/L)	References
Glucose	Naringenin	*Y. lipolytica*	252.4	[130]
Glucose	Eriodictyol	*Y. lipolytica*	134.2	[130]
Glucose	Taxifolin	*Y. lipolytica*	110.5	[130]
Genistein	4′-*O*-methyl genistein	*E. coli*	46.81	[147]
Daidzein	4′-*O*-methyl daidzein	*E. coli*	102.88	[147]
Naringenin	Kaempferol	*E. coli*	1184.2	[153]
Naringenin	Astragalin	*E. coli*	1738.5	[153]
Hesperetin	Hesperetin-3′-*O*-rhamnoside	*E. coli*	2400	[151]
Quercetin	Quercitrin	*E. coli*	4300	[151]
Kaempferol	Afzelin	*E. coli*	1900	[151]
Catechin and Glucose	Cyanidin 3-*O*-glucoside	*E. coli*	439	[154]
Tyrosine	Scutellarein	*E. coli*	106.2	[155]
Phenylalanine	Baicalein	*E. coli*	23.6	[155]
Naringenin	Genistein	*E. coli*	35	[128]
p-Coumaric acid	Genistein	*E. coli*	18.6	[128]
Tyrosine and malonate	Naringenin	*E. coli*	191.9	[177]
Apigenin	Isovitexin	*E. coli*	3772	[178]
Luteolin	Isoorientin	*E. coli*	3820	[178]
Catechin	Cyanidin 3-*O*-glucoside	*C. glutamicum*	40	[139]
Sucrose and glycerol	Naringenin	*S. cerevisiae*	28.68	[18]
Glucose	Kaempferol	*S. cerevisiae*	86	[159]
Xylose	p-Coumaric acid	*S. cerevisiae*	242	[2]
Glucose	Delphinidin 3-*O*-glucoside	*S. cerevisiae*	1.86	[179]
Glucose	Cyanidin 3-*O*-glucoside	*S. cerevisiae*	1.55	[179]
Glucose	Pelargonidin 3-*O*-glucoside	*S. cerevisiae*	0.85	[179]
Glucose	Icaritin	*S. cerevisiae*	7.2	[180]
Glucose	Taxifolin	*S. cerevisiae*	336.8	[181]
Green tea	Anthocyanin	*L. lactis*	1.5	[157]

**Table 5 metabolites-13-00124-t005:** Co-culture approach for the microbial production of flavonoids.

Co-Culture Strains	Substrate	Product	Titer (mg/L)	References
*E. coli*–*E. coli* coculture	Glucose	Sakuranetin	29.7	[182]
*E. coli*–*E. coli* coculture	(+)-Catechin and glucose	Pyranocyanidin-3-*O*-glucoside-catechol	13	[185]
*E. coli*–*E. coli* coculture	(+)-Catechin, glucose, and tyrosine	Pyranocyanidin-3-*O*-glucoside-phenol	19.5	[185]
*E. coli*–*E. coli* coculture	p-Coumaric acid	Apigetrin	16.6	[183]
*E. coli*–*E. coli* coculture	Apigenin and luteolin	Orientin	7090	[187]
*E. coli*–*E. coli* coculture	Apigenin and luteolin	Vitexin	5050	[187]
*E. coli*–*S. cerevisiae* coculture	Glucose	Icaritin	19.7	[180]
*E. coli*–*S. cerevisiae* coculture	Xylose	Naringenin	21.16	[188]
*S. cerevisiae*–*S. cerevisiae* coculture	Naringenin	Delphinidin	26.1	[20]
*S. cerevisiae*–*S. cerevisiae* coculture	p-Coumaric acid	Naringenin	18.5	[189]

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
