# Peer review of "Flavonoid Production: Current Trends in Plant Metabolic Engineering and De Novo Microbial Production"

_metabolites, 2023, doi:10.3390/metabo13010124_

Round 1

Reviewer 1 Report

Title: The focus of the review was on production rather than biosynthesis. It should be changed accordingly.

Abstract: Lines 21 & 22: There is not enough coverage of the flavonoids in the review that are pharmaceutically important but not industrially or vice versa.

Introduction:

Line: 49: What do the authors mean from respectively?

Line 54: How is the extraction yield related to the listed factors? Please re-frame the sentence. It could be due to their low concentration or losses during the extraction.

Line: 61: All of these constraints .... . The message is not clear. Revise the sentence.

Line: 67: Editing of .... . Redundancy and unclear terms in the sentence.

Line 72: What kind of emission? What waste is being talked about?

2. Flavonoid Biosynthetic Pathways in Plants, and 3. An Overview of Key Enzymes involved in Biosynthesis of Flavonoids:

In these sections, the authors did not provide any new information on the biosynthesis challenges, especially on the regulation at molecular, enzymatic and cellular levels. The pathway details have been summarized in many recent review articles. The table is packed with trivial details such as EC number, class, family, etc. The flavonoid synthesis pathways have also been graphically represented later in the review. Sections 2 and 3 do not add value to the review in the present format. The authors should find ways to avoid duplication in biosynthetic pathways. The information must be succinct, and the details should be just enough to support other section discussions.

4. Metabolic Engineering of the Flavonoid Pathways:

4.1. Metabolic Engineering in Plants:

Line 210: The introduction .... . Out of context. Please explain what you want to say.

Lines 214-216: Secondly, the .... . Not sure if this is a strategy or just stating the facts about plant metabolism. Rectify it accordingly.

Lines 216-217: What do the authors mean from metabolic flux? Please elaborate.

Line: 240: Anthocyanins .... . This is a random insertion of a sentence.

Lines 259-262: The statement undermines the challenges of increased flavonoid production in plants.

Table 2: Repetitive information creates redundancy, which is amply found in this table. What purpose does it serve to say, Ag-mediated, overexpression, again and again? It could be said in a single sentence somewhere in the text.

Table 3. Please provide the rationale why this table is needed. Explain the value of information in this table. Otherwise, it could be deleted.

General comment: The overexpression of genes is not synonymous with the up-regulation of encoding enzymes. The authors should be careful while discussing this aspect. There needs to be some discussion on the differential regulation of plant and microbial genes involved in flavonoid biosynthesis. If this aspect has been exploited, it should be elaborated.

4.2. Metabolic Engineering in Microbes:

Line 274: Variation of what?

Line 279: Explain how the flavonoids synthesized by microbes are highly pure.

Line 282: The figure lacks clarity. See more comments elsewhere.

Lines: 288-290: The sentence is unclear. Please re-frame it.

Lines 300-303: The authors should provide examples or the progress made so far on all those points. 

Line 312: What does it mean, via demonstration? Elaborate.

Figure 3. The conversion of glucose to acetyl CoA has been shown through a direct arrow outside glycolysis. Is there a different pathway? If so, then show the steps. The figure is very complex and difficult to comprehend. It does not provide a good view of different pathways. In one section of the figure, it depicts methanol as the substrate. Provide an example in the text. Similarly, xylose as a substrate seems to be feeding into the PPP. Please provide details on where it is happening. The figure can be broken into sub-figures to provide clarity. The figure requires some creativity to accommodate all aspects.

Lines 336-341: A complex sentence that lacks clarity. Also, it is less convincing that chemical synthesis and plant cultures produce the compounds with low purity compared with microorganisms.

Lines 362-364: A random insertion of a sentence that talks about the advantages of hydroxylated flavonoids while discussing the microbial challenges of their production.

Line: 370: What was the level of improvement over wild-type strains?

Line 529: Do you mean a precursor of PEP?

Line 543: It is unclear if you meant to say these metabolites as precursors or some other metabolites as precursors of E4P and PEP.

Lines 554-558: There is a sentence structure issue to convey the correct message.

Table 4: The titre values are displayed in an inconsistent way. In some places, it's up to two decimals places of mg and other in places, the values are one decimal place. 

General comment: There is a slight difference between the terms 'substrate' and 'precursor'. In metabolism, a metabolite could be a substrate as well as a precursor. The terms have not been used keeping this distinction in mind causing confusion in many places.

5. Microbial Co-Culture Strategy for Flavonoids Biosynthesis:

Figure 4: What do the double arrows around pathways represent? Do all the scenarios (a,b,c) show the same substrate, intermediates and products?

Line 637: What does 'substantially elevated' refer to?

6. Conclusion

The conclusion should not be a copy of the introduction. It should be a succinct summary of the important details of the review with a visionary message.

Overall comment: The authors have compiled valuable information on the important flavonoids that are revolutionlizing the food, ornamental and pharmaceutical industries. They have discussed plant and microbial perspectives on their production. The manuscript needs to be refined, removing information that only inflates the text but offers little value. Close attention is required to address the redundancy and monotonic style of writing in certain sections. On multiple occasions, the verb did not match the subject, spelling mistakes, and there were some other grammar issues. More than one style has been used to quote the references in the text. They should adopt a style according to the journal guidelines. Once the manuscript is revised following the suggestions, it should provide the reader with its worth.

Author Response

Response to Reviewer comments

Reviewer 1

Title-The focus of the review was on production rather than biosynthesis.

Response- Manuscript title has been edited.

Lines 21 & 22: There is not enough coverage of the flavonoids in the review that are pharmaceutically important but not industrially or vice versa.

Response- Line has been corrected.

Line 49: What do the authors mean from respectively?

Response- Line has been edited

Line 54: How is the extraction yield related to the listed factors? Please re-frame the sentence. It could be due to their low concentration or losses during the extraction.

Response- Sentence has been reframed according to comment.

Line: 61: All of these constraints .... . The message is not clear. Revise the sentence.

Response- Sentence structure is edited to elaborate the meaning.

Line: 67: Editing of .... . Redundancy and unclear terms in the sentence.

Response- Line has been edited.

Line 72: What kind of emission? What waste is being talked about?

Response- Kinds of waste emissions have been listed.

  1. Flavonoid Biosynthetic Pathways in Plants, and 3. An Overview of Key Enzymes involved in Biosynthesis of Flavonoids:

In these sections, the authors did not provide any new information on the biosynthesis challenges, especially on the regulation at molecular, enzymatic and cellular levels. The pathway details have been summarized in many recent review articles. The table is packed with trivial details such as EC number, class, family, etc. The flavonoid synthesis pathways have also been graphically represented later in the review. Sections 2 and 3 do not add value to the review in the present format. The authors should find ways to avoid duplication in biosynthetic pathways. The information must be succinct, and the details should be just enough to support other section discussions.

Response- 

  • A new paragraph focusing on the biosynthetic challenges has been incorporated in the ‘Flavonoid Biosynthetic Pathways in Plants’ section.
  • Table 2 has been edited and EC number column has been removed
  • Section 2 and 3 have been edited and minor corrections have been made.
  1. Metabolic Engineering of the Flavonoid Pathways:

4.1. Metabolic Engineering in Plants:

Line 210: The introduction .... . Out of context. Please explain what you want to say.

Response- The introduction provides some background on plant metabolic engineering and purposes, the sentence structure has been edited to better convey the meaning.

Lines 214-216: Secondly, the .... . Not sure if this is a strategy or just stating the facts about plant metabolism. Rectify it accordingly.

Response- The paragraph states different challenges in a first, second, and third manner. Sentence structure has been edited to elaborate the meaning.

Lines 216-217: What do the authors mean from metabolic flux? Please elaborate.

Response- Required changes have been made.

Line: 240: Anthocyanins .... . This is a random insertion of a sentence.

Response- The sentence has been removed from the paragraph.

Lines 259-262: The statement undermines the challenges of increased flavonoid production in plants.

Response- Sentence structure has been edited to elaborate

Table 2: Repetitive information creates redundancy, which is amply found in this table. What purpose does it serve to say, Ag-mediated, overexpression, again and again? It could be said in a single sentence somewhere in the text.

Response- Table 2 has been edited. Two columns have been removed and Ag-Mediated overexpression has now been included in the paragraph section ‘Metabolic Engineering in Plants’

Table 3. Please provide the rationale why this table is needed. Explain the value of information in this table. Otherwise, it could be deleted.

Response- Table 3 depicts prospects for potential targets of metabolic engineering approaches. The aim to identify novel gene candidates which have yet to be exploited or require more research.

General comment: The overexpression of genes is not synonymous with the up-regulation of encoding enzymes. The authors should be careful while discussing this aspect. There needs to be some discussion on the differential regulation of plant and microbial genes involved in flavonoid biosynthesis. If this aspect has been exploited, it should be elaborated.

Response- The manuscript has been revised to outline and correct general mistakes.

4.2. Metabolic Engineering in Microbes:

Line 274: Variation of what?

Response- It has been edited to hard-to-culture plant species.

Line 279: Explain how the flavonoids synthesized by microbes are highly pure.

Response- The sentence has been changed. However, the comment regarding the purity of flavonoids produced by microbes is addressed elsewhere.

Line 282: The figure lacks clarity. See more comments elsewhere.

Response- The comments have been addressed elsewhere.

Lines: 288-290: The sentence is unclear. Please re-frame it.

Response- The sentence is reframed.

Lines 300-303: The authors should provide examples or the progress made so far on all those points.

Response- The relevant review articles which discussed them in greater detail have been cited.

Line 312: What does it mean, via demonstration? Elaborate.

Response- The sentence has been modified.

Figure 3. The conversion of glucose to acetyl CoA has been shown through a direct arrow outside glycolysis. Is there a different pathway? If so, then show the steps. The figure is very complex and difficult to comprehend. It does not provide a good view of different pathways. In one section of the figure, it depicts methanol as the substrate. Provide an example in the text. Similarly, xylose as a substrate seems to be feeding into the PPP. Please provide details on where it is happening. The figure can be broken into sub-figures to provide clarity. The figure requires some creativity to accommodate all aspects.

Response- The glucose to acetyle CoA pathway has been corrected in the figure. Additionally, a study regarding the utilization of methanol as a substrate has also been cited in the text. The xylose pathway has been removed due to its complexity which can be represented in the figure. Still if the figure isn’t upto the standard please tell us whether we should keep it or remove it.

Lines 336-341: A complex sentence that lacks clarity. Also, it is less convincing that chemical synthesis and plant cultures produce the compounds with low purity compared with microorganisms.

Response- The sentence structure has been revised. Also, the term "plant cultures" has been deleted, although when compared to chemical synthesis, microorganisms produce pure compounds. First, the use of microbes eliminates the use of heavy metals, organic solvents, and strong acids and bases, allowing the synthetic process to adopt a more ecologically friendly path. Second, enzymes typically have a relatively high substrate specificity, which aids in the reduction of byproduct formation. Finally, certain molecules with complicated structures already have natural synthesis pathways, but building chemical synthetic routes for these complex chemicals is extremely challenging.

Du, J., Shao, Z., & Zhao, H. (2011). Engineering microbial factories for synthesis of value-added products. Journal of Industrial Microbiology and Biotechnology38(8), 873-890.

Lines 362-364: A random insertion of a sentence that talks about the advantages of hydroxylated flavonoids while discussing the microbial challenges of their production.

Response- The sentence has been removed.

Line: 370: What was the level of improvement over wild-type strains?

Response- Improvements in the production of naringenin, eriodictyol, and taxifolin have been mentioned.

Line 529: Do you mean a precursor of PEP?

Response- Yes, but there is another precursor (E4P) which is discussed later in the paragraph so we didn’t mention PEP in the first sentence and wrote it generally.

Line 543: It is unclear if you meant to say these metabolites as precursors or some other metabolites as precursors of E4P and PEP.

Response- The sentence has been edited to convey the correct meaning. Basically, we meant to say precursors (E4P and PEP) as both of them are required for the production of L-DOPA. The following link explains the process.

https://microbialcellfactories.biomedcentral.com/articles/10.1186/s12934-019-1122-0/figures/4

Lines 554-558: There is a sentence structure issue to convey the correct message.

Response- The sentence structure has been edited and improved.

Table 4: The titre values are displayed in an inconsistent way. In some places, it's up to two decimals places of mg and other in places, the values are one decimal place. 

Response- Corrections have been done in Table 4.

General comment: There is a slight difference between the terms 'substrate' and 'precursor'. In metabolism, a metabolite could be a substrate as well as a precursor. The terms have not been used keeping this distinction in mind causing confusion in many places.

  1. Microbial Co-Culture Strategy for Flavonoids Biosynthesis:

Figure 4: What do the double arrows around pathways represent? Do all the scenarios (a,b,c) show the same substrate, intermediates and products?

Response- The figure 4 has been edited. The single arrow above each pathway represents the promoter. Secondly, all the scenarios shows the same substrate, intermediates, and products. Basically, it explains the ways we can use different microbes in a co-culture system for the production of metabolites.

Line 637: What does 'substantially elevated' refer to?

Response- ‘Substantially elevated’ refered to the production of sakuranetin from the co-culture system compared to the monocultured approach. It explains that the sakuranetic production was comparatively higher in the co-cultured system.

  1. Conclusion

The conclusion should not be a copy of the introduction. It should be a succinct summary of the important details of the review with a visionary message

Response- The conclusion has been modified.

Additional:

  • Reference list has been updated (1 removed- 186 total)
  • All in-text citations have been edited according to journal guidelines
  • Grammatical errors and mistakes have been corrected.

Reviewer 2 Report

This review entitled “Flavonoid Biosynthesis: Current Trends in Plant Metabolic Engineering and De novo Microbial Production” submitted by Tariq and all., discussed about the challenges of flavonoid biosynthesis and present recent solutions to circumvent critical points sing microbial engineering. Authors clearly explained flavonoid biosynthesis in plants and the different approaches that can be used for flavonoids production in plants and in microbial hosts, such as co-culture systems. They listed enzymes involved in flavonoid biosynthesis and summarized their literature findings in clear tables.

The manuscript is clear and well written. I would therefore recommend its acceptance for publication after minor revisions and corrections..

Line 11:

Flavonoids are not commonly only found in edible plants and fruits but in all plants.
I would recommend to rephrase it by saying something like:

“Recent discoveries have revealed that the benefit of fruits, vegetables on our health, and the therapeutic potential of medicinal plants are based on the presence of various bioactive natural products, including a large proportion of flavonoids”.

Line 99:

What is the meaning of the big red cross on the image for “Large scale production”

Line 105:

Please correct “tp”

Author Response

Response to Reviewer comments

Reviewer 2

Line 11: Flavonoids are not commonly only found in edible plants and fruits but in all plants.
I would recommend rephrasing it:

Response- Required rephrasing has been done and new lines have been included.

Line 99: What is the meaning of the big red cross on the image for “Large scale production”

Response- The big red cross in the Figure 1 shows that in case of chemical synthesis and plant extraction, there are many challenges to scale up those processes and those challenges can be overcome if microbes are used for the production of flavonoids.

Line 105: Please correct “tp”

Response- Word has been corrected ‘To’
